# Broadband Transition from Rectangular Waveguide to Groove Gap Waveguide for mm-Wave Contactless Connections

**Zihao Liu [1]** , **Xiaohe Cheng [1,\*]**, **Yuan Yao [1]**, **Tao Yu [2]**, **Junsheng Yu [1]** and **Xiaodong Chen [3]**

1   School of Electronic Engineering, Beijing University of Posts and Telecommunications, Beijing 100876, China; Liu.zihao@bupt.edu.cn (Z.L.); yaoy@bupt.edu.cn (Y.Y.); jsyu@bupt.edu.cn (J.Y.)
2   Department of Electrical and Electronic Engineering, Tokyo Institute of Technology, Tokyo 152-8552, Japan; yutao@mobile.ee.titech.ac.jp
3   School of Electronic Engineering and Computer Science, Queen Mary University of London, London E1 4NS, UK; xiaodong.chen@qmul.ac.uk
*   Correspondence: xiaohec@bupt.edu.cn

**Abstract:** In this paper, the authors present a broadband transition from the standard WR-10 rectangular waveguide (RW) to a groove gap waveguide (GGW) in the W-band. The transition structure is based on electromagnetic band gap (EBG) technology where two EBG units are used, which are responsible for the transition and forming the transmission line. Metal pins in the E-plane together with the back surface of the transmission line create a forbidden band, which prevents power leakage between the connecting parts. Small air gaps will not harm the transition performance according to the simulation, which means it has a better tolerance of manufacturing and assembly errors and, thus, has advantages for mm-wave contactless connections. A back-to-back transition prototype was designed, fabricated and measured. The length of the GGW is 39.6 mm. The measured $|S_{11}|$ is better than −13 dB and the measured $|S_{21}|$ is better than −0.6 dB over 76.4–109.1 GHz, covering a bandwidth of 35.3%.

**Keywords:** contactless connections; groove gap waveguide; mm-wave; waveguide transition

## 1. Introduction

Fifth-generation (5G) wireless networks will be able to achieve an increase in capacity of 1000-fold or more over 4G and the use of mm-wave bands is a key enabler to making 5G able to achieve such increase in capacity because mm-wave bands can offer more than 1 GHz of continuous spectrum and are able to provide data rates of multi-Gigabit-per-second (Gbps) [1].

In recent years, research has been conducted to enhance the performance of 5G mm-wave communication systems. In [2], the authors proposed a low-complexity and accurate mm-wave position estimation approach. In [3], the design of hybrid analog or digital precoder and combiner was proposed to reduce power consumption and hardware complexity and achieve a better performance for mm-wave multiple-input-multiple-output (MIMO) systems. For this work, the authors focus on building a more flexible hardware platform for mm-wave communication systems.

A gap waveguide is formed by a tiny gap between two parallel metal plates and has no requirements for electrical conducting joints between the two plates [4]. In the case of a groove gap waveguide (GGW), the field propagates through the air channel between two sections of pins [5].

Up to now, various passive components based on the GGW, such as filters [6], orthomode transducers [7], diplexers [8], couplers [9], waveguide flanges [10,11], twisted waveguide [12] and antenna arrays [13], have been proposed. To match the interfaces of other equipment, several types of

transition structures have been proposed correspondingly. The GGW-to-microstrip line [14,15] and the ridge gap waveguide (RGW)-to-coplanar waveguide [16] for monolithic microwave integrated circuit (MMIC) integration were proposed. In [17], a transition from coaxial-to-RGW with a 4:1 bandwidth was designed to cover from Ku- to Ka-band. In [18], a vertical transition from a rectangular waveguide (RW) to a GGW with step-impedance matching structure was presented at the V-band. In [19], a vertical RW-to-GGW transition including a power divider is designed to match the corporate feeding network array.

　　　Meanwhile, the horizontal transition from an RW to a GGW is also needed for filters, end-fire antennas, couplers, etc. Under full electrical contact conditions, the RW can match to the GGW with the same aperture in horizontal direction. However, the performance of this transition structure is very sensitive at interconnection between the RW and GGW. A few micrometers gaps and misalignments would cause a significant difference in S-parameter at the millimeter-wave band. Furthermore, when applied to antennas, they will lead to deterioration in the radiation efficiency. In recent years, the modified contactless RW flange was proposed in [10–12], which is used to connect two RWs with the same waveguide aperture dimension.

　　　To address the challenge, in this paper, a broadband horizontal transition with a contactless structure from an RW to a GGW is first proposed. The transition structure in the E-plane is based on electromagnetic band gap (EBG) technology, which is developed from the modified contactless RW flange [10–12]. Finally, a W-band prototype was designed, fabricated and measured.

## 2. Design of the Transition

　　　Figure 1 shows the 3D view of the back-to-back transition. Based on the studies of the cut-off bandwidth of EBG structure when using different pin dimensions [20], the dispersion diagram of EBG units #1 and #2 was simulated in HFSS Eigenmode and is shown in Figure 2. Dimensions of the units are summarized in Table 1. The cut-off bandwidth for both of the two units can cover the whole W-band.

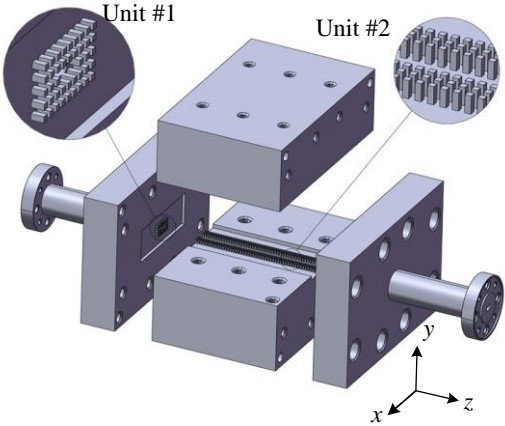

**Figure 1.** 3D view of the back-to-back transition. The left circle contains gap waveguide unit #1 while the right circle refers to gap waveguide unit #2.

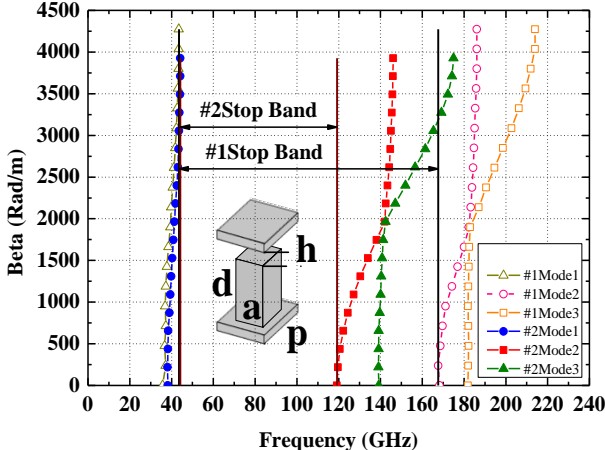

**Figure 2.** Dispersion diagram of electromagnetic band gap (EBG) unit #1 and #2.

**Table 1.** Dimension of the EBG units (unit: millimeter).

| $a_1$ | $p_1$ | $d_1$ | $h_1$ | $a_2$ | $p_2$ | $d_2$ | $h_2$ |
|-------|-------|-------|-------|-------|-------|-------|-------|
| 0.4 | 0.735 | 0.84 | 0.02 | 0.4 | 0.8 | 1.15 | 0.05 |

In the $xy$ plane, the RW is surrounded by EBG unit #1 and the structure is symmetrical about the $x_-$ and $y_-$ axis, as presented in Figure 3. The EBG units marked by black lines are EBG unit #1 and their height in $z\_direction$ is represented by the variable $d_1$. The six EBG units among the short side of WR-10 RW are marked by red lines and their height in $z\_direction$ is represented by the variable $z_1$. In the final fabricated version, $d_1$ and $z_1$ are both equal to 0.84 mm, and so, in the $xy$ plane, all the EBG units are unit #1. The discussion and optimization of $d_1$ and $z_1$ are presented in Section 3. For the optimization of $z_1$, $d_1$ equals 0.84 mm. For the optimization of $d_1$, the EBG units marked by red lines are not included in the analysis.

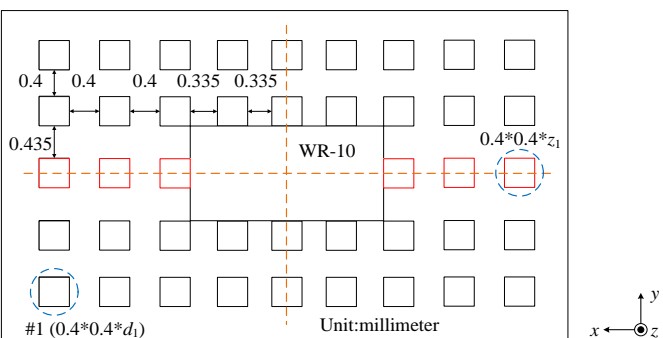

**Figure 3.** Configuration of the transition in the $xy$ plane.

In the $xz$ plane, three periods of EBG unit #2 form the GGW. In this work, its total height is 1.2 mm to match that of WR-10 and the width of the groove in the GGW is 2.54 mm. The transition part is attached to the center of the GGW.

## 3. Parameter Analysis

The direct connection of the GGW and the RW [15] shows good performance in simulation results, but in practice, the two parts cannot be connected so tightly after fabrication due to unavoidable manufacturing errors and misalignments. To evaluate the influence of the gap between the GGW and the RW, a GGW-to-RW transition structure with an air gap of *gap_z* on the PORT1 side is modeled in Figure 4. It should be noted that the dimensions of the GGW are kept the same with unit #2 and

the length of the GGW is set to be 39.6 mm. The transmission coefficients |S$_{21}$| by different values of *gap_z* are illustrated in Figure 5, which show that the transmission performance has a very significant deterioration when the air gap is greater than 0.02 mm, which is a normal machining error in practical milling machines.

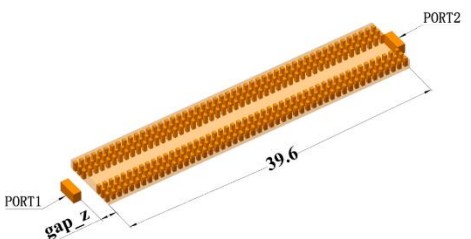

**Figure 4.** Simulation model for the study of the influence of *gap_z* on transition performance.

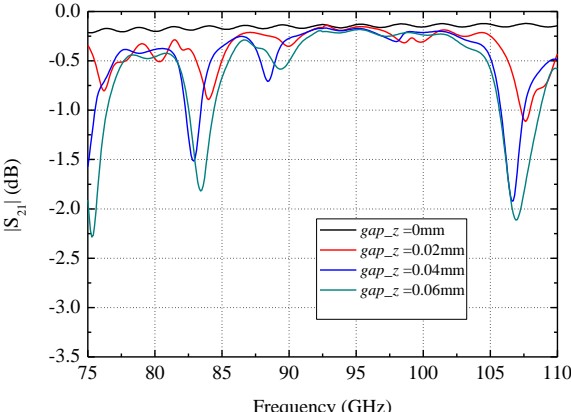

**Figure 5.** Simulation results of variation in different length of air gaps (*gap_z*).

In order to further analyze the power leakage phenomenon, a single transition model of the RW to a GGW is used in Figure 6a. The air gap is 0.2 mm in length. Figure 6a,b is the electric field distribution and magnetic field intensity of the air gap plane at 110 GHz, respectively. The fundamental mode of the WR-10 RW is the TE$_{10}$ mode and both the transverse electric field and magnetic field follow the sin-function distribution. Based on the Poynting theorem, which is

$$S = E \times H \tag{1}$$

the short side of the RW has low power density, so the power leakage near this area is little. Among the long side of the RW, the power density is high in the center area, and as a result of its vertical polarization, the electromagnetic wave will propagate in the vertical direction. Figure 6a,b illustrate that the in the vertical direction, both the electric field density and magnetic field intensity are high, making the vertical direction the main power leakage direction.

Therefore, EBG units in the E-plane mainly focus on preventing power leakage in the vertical direction due to the effects of wave polarization. The units among the short side will not have an obvious impact on the transition performance. In the original design edition, the E-plane EBG units #1 only contain the units marked by black lines in Figure 3 in order to prevent vertical power leakage. In Figure 3, the EBG units among the short side of the RW have different boundary conditions compared with other units in the E-plane. Unlike the traditional EBG units backed by two Perfect Electric Conductor (PEC) plates, one side of these units is backed by the vertically placed EBG unit #2. Due to the difference in the boundary conditions, $z_1$ and $d_1$ need to be optimized. In the back-to-back transition model, $z_1$ ranges from 0.1 to 0.86 mm.

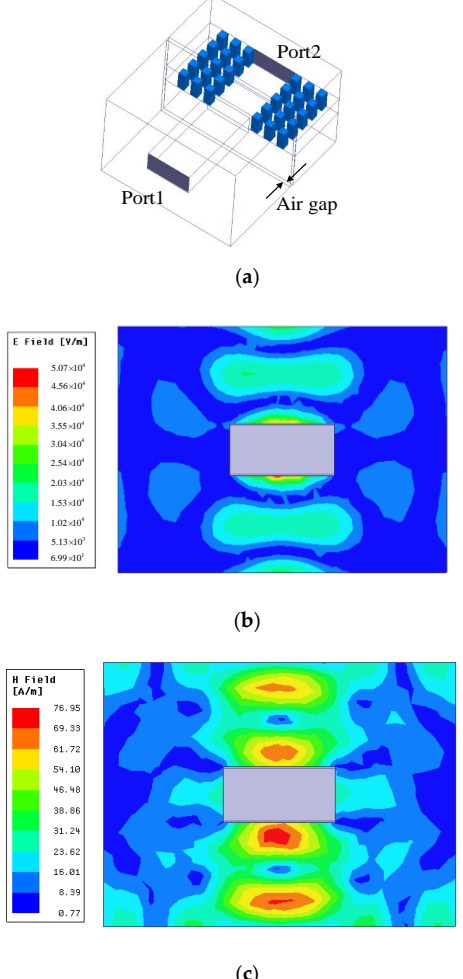

**Figure 6.** Analysis of the effects of wave polarization. (**a**) A single rectangular waveguide (RW)-to-groove gap waveguide (GGW) transition model; (**b**) electric field distribution of the air gap plane at 110 GHz; (**c**) magnetic field intensity of the air gap plane at 110 GHz.

The lower border is set to be 0.1 mm considering the fabrication difficulties and the upper border is the total height of EBG unit #1. The simulated result in Figure 7 matches the conjecture, $z_1$ has little effect on the transition performance and there will not be a power leakage even with small height of the pins. The total height of EBG unit #1 is 0.86 mm, if $z_1$ is set to be 0.86 mm; the EBG unit #1 among the short side of the RW would be fully connected to EBG unit #2, which requires a high flatness of metals. Our machining accuracy is 0.01 mm; due to fabrication concern, the air gap $h_1$ should be equal to or greater than 0.02 mm to have an allowance. For comprehensive consideration, the authors chose $z_1 = 0.84$ mm.

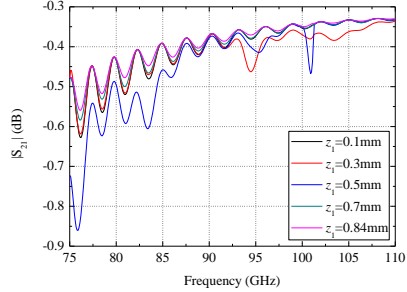

**Figure 7.** Transition performance given by different values of $z_1$.

The height of EBG unit #1 $d_1$ will have an influence on the stopband for pins and then affect the performance of the transition. In addition, a smaller $d_1$ can achieve a better impedance match within the stopband for pins. The $|S_{21}|$ with different values of $d_1$ are presented in Figure 8. The performance when $d_1 = 0.44$ mm is better than $d_1 = 0.84$ mm. However, a smaller value of $d_1$ will result in a narrower stopband for EBG units as well as a smaller error tolerance of air gap. Therefore, the authors chose $d_1 = 0.84$ mm for a better tolerance of the manufacturing error.

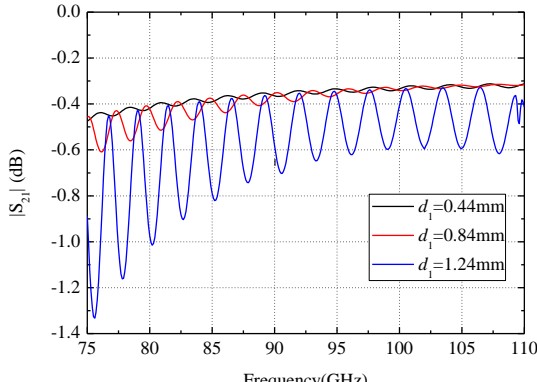

**Figure 8.** Transition performance given by different values of $d_1$.

The EBG unit #1 acts as an artificial magnetic conductor (AMC) surface. Therefore, electromagnetic waves cannot be leaked from the gap within the stopband of pins. On the other hand, the stopband for pins has a higher tolerance for the gap. As shown in Figure 9, $h_1$ represents the air gap on the PORT1 side, with an initial value of 0.02 mm, and the other dimensions of the transition structure remain the same as in Figure 4. The simulation results of the proposed transition in Figure 10 illustrate that even with an air gap as large as 0.06 mm, the transition performance is only slightly deteriorated and still well within an acceptable range, which means that the stopband for pins still covers the W-band with an air gap of 0.06 mm. Simulation results given in Figure 5 prove that the direct connection method has a poor tolerance of air gaps; when *gap_z* equals or is greater than 0.02 mm, the $|S_{21}|$ value is worse than −1 dB. Meanwhile, utilizing the presented E-plane EBG structure, even with an air gap of 0.06 mm, the $|S_{21}|$ value is better than −0.6 dB, as is shown in Figure 10. Therefore, by adding the pins at the interconnection between the RW and GGW, a comparatively much better tolerance of air gaps can be achieved.

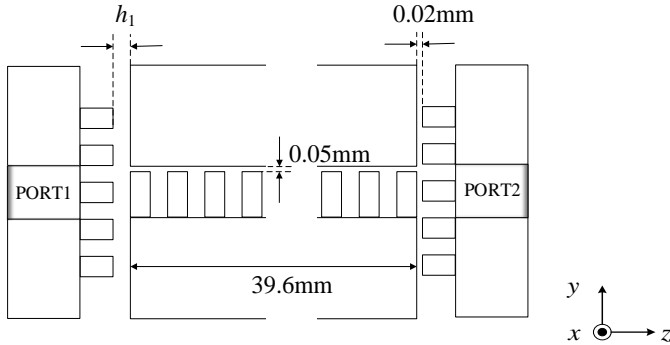

**Figure 9.** Side-view of the transition in the *yz* plane. $h_1$ is the distance between unit#1 and #2 in the *z_* direction on PORT1 side.

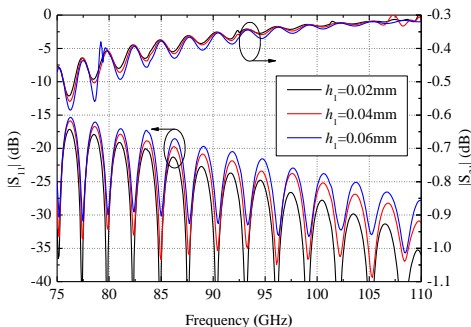

**Figure 10.** Performance of the proposed transition given by different values of $h_1$.

As mentioned above, the transition structure has good tolerance of air gaps in the $z\_$ direction. In this part, tolerance analysis is performed to evaluate the influence of the misalignment of the transition part in the $x\_$ or $y\_$ direction. In Figure 11a, the red line indicates the original correct position of the transition part and the dashed line indicates the position of it after misalignment. Since the transition part is symmetrical about the $x\_$ and $y\_$ axis, misalignment on one side each is studied. The results in Figure 11b,c illustrates that the appearance of misalignment in the $x\_$ or $y\_$ direction within 0.2 mm is acceptable, which is much greater than the assembly errors.

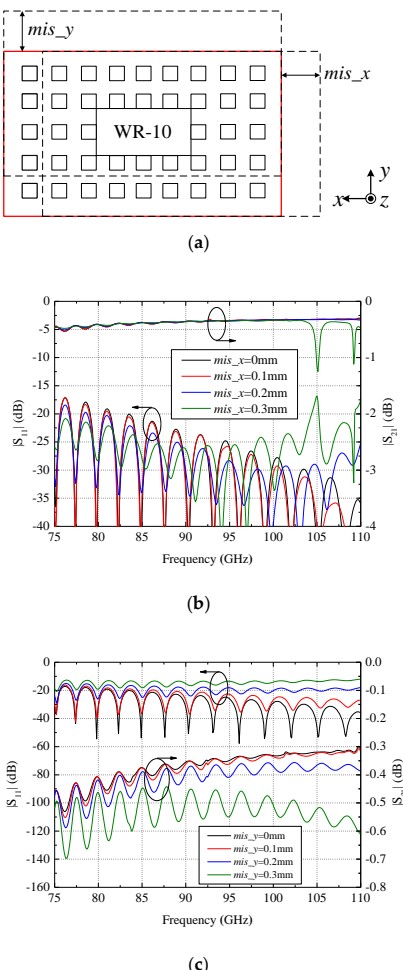

**Figure 11.** Tolerance analysis of the misalignment in the $x\_$ or $y\_$ direction. (**a**) Schematic diagram of the transition part in the $xy$ plane; (**b**) simulation results given by different misalignments in the $x\_$ direction; (**c**) simulation results given by different misalignments in the $y\_$ direction.

## 4. Design Methods

Based on the design of the transition and parameter analysis, the design method of the transition is discussed in this section.

Firstly, based on the purpose of a working frequency band, the size of the RW and the width of the groove in the GGW can be confirmed. On the design of EBG unit #2, to achieve good impedance matching, the total height of the units should be around that of the RW. The stopband of the units should cover the W-band.

On the design of EBG unit #1, the stopband of the units should cover the W-band as well. The EBG units should be placed along the long side of the RW, forming an artificial magnetic conductor (AMC) surface to confine the electromagnetic wave inside the RW area. Four EBG units should be placed at the corners of the RW for better positioning purposes. Then, several EBG units should be placed along the long side of the RW on each side. Two rows of EBG units on each long side of the RW are placed to prevent power leakage in this work. Finally, one row of EBG units is placed along the short side of the RW. It should be noted that each EBG unit #1 should have a similar period to create a near stopband.

## 5. Results and Discussion

A back-to-back transition prototype was fabricated by milling and discharge in copper and measured by a Ceyear3672D vector network analyzer with two 3643P W-band extenders. A photograph of the prototype is given in Figure 12. The fabricated prototype contains four pieces. Pieces 1 and 4 are of the same structure—both have a WR-10 flange on the outer side and an EBG unit #1 on the inner side. Piece 3 contains EBG unit #2 and Piece 2 is a metal block and act as the top metal plate for EBG unit #2. The rectangular waveguides are 40 mm in length and the output ports are directly connected to standard UG-387 flanges. The GGW in the middle is 39.6 mm in length. All four parts of the prototype are connected by M3 Screws. Simulations were done by HFSS. The modeling size of the RWs is 2.54 × 1.27 mm and they have a length of 40 mm; two exciting wave ports are on the cross sections of the RWs. The whole structure is surrounded by a 1.5-mm larger air box with radiation boundary. All the dimensions of the simulated models conform to the parameters given in Sections 2 and 3.

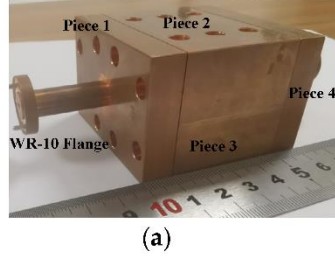
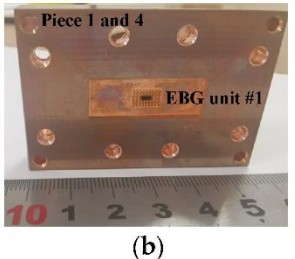
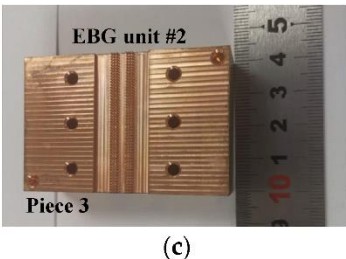

(**a**)　　　　　　　　　　(**b**)　　　　　　　　　　(**c**)

**Figure 12.** Photographs of the proposed transition prototype. (**a**) Overall appearance of the fabricated prototype; (**b**) Pieces 1 and 4 with EBG unit #1; (**c**) Piece 3 with EBG unit #2.

As illustrated in Figure 13, the simulated $|S_{11}|$ is better than −15 dB and $|S_{21}|$ ranges from −0.3 to −0.54 dB over the whole W-band. The measured $|S_{11}|$ is better than −13 dB over the whole W-band. The $|S_{11}|$ measurement result shows stable return loss over the whole W-band but does not perfectly agree with the simulated result, especially at 90 GHz and above. Measured $|S_{21}|$ ranges from −0.33 to −0.4 dB over 85.5–97.4 GHz perfectly agreed with the simulated results. Beyond this frequency range, however, oscillation of the $|S_{21}|$ waveform increases as the frequency approaches 75 GHz or 110 GHz. Over 75–76.4 GHz and 109.1–110 GHz, oscillation of the $|S_{21}|$ waveform becomes more obvious and the measured $|S_{21}|$ shows a fast decrease. As a whole, the measured $|S_{21}|$ is better than −0.6 dB over 76.4–109.1 GHz, covering a bandwidth of 35.3%, and the measured $|S_{21}|$ agrees well with the simulated result in this frequency band. The oscillation of the waveform is mainly introduced by periodic EBG structures and testing errors.

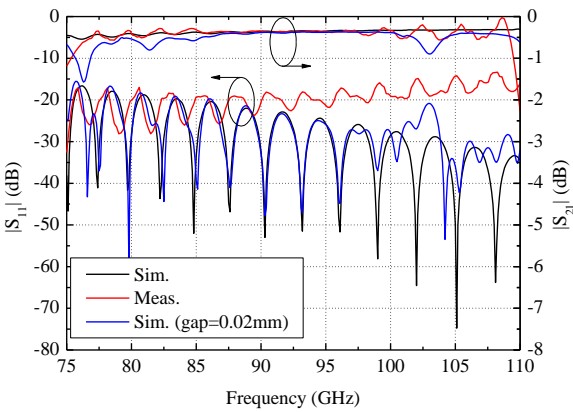

**Figure 13.** Simulation and measured results of the fabricated prototype.

As mentioned in Section 3, if the transition parts cannot connect tightly enough, there will be a power leakage. The structure in this work prevents leakage between the RW and GGW but air gaps could still appear at the interconnection between the UG-387 flanges during measurement. To estimate the influence of misalignment or air gaps between the testing flange and the flange on our prototype, during simulation, two additional $2.54 \times 1.27$-mm RWs representing the testing flanges are attached to the original simulation model. The exciting ports are on these additional RWs. Simulations were done by HFSS and the results show that misalignments of the WR-10 flanges in the $x\_$ or $y\_$ directions have little effect on the transition, while air gaps in the $z\_$ direction influence the transition performance to a large extent. An air gap of 0.02 mm in the $z\_$ direction between the flanges was used to approximate the testing results in Figure 13.

Table 2 below lists some conventional transitions from a gap waveguide to other transmission lines as a comparison. RL refers to return loss and IL refers to insertion loss.

**Table 2.** Gap waveguide transitions and performance.

| Ref. | Transition from | Transition to | Frequency (GHz) | RL (Measured) | IL (Measured) |
|---|---|---|---|---|---|
| [14] | microstrip line | GGW | 90–99.5 | >12 dB | <5 dB |
| [15] | microstrip line | GGW | 83.9–109 | >10 dB | <4 dB |
| [17] | coaxial | RGW | 12.8–40 | >13 dB | 0.8 dB (average) |
| [18] | RW | GGW | 50–75 | >13 dB | <0.4 dB |
| Proposed | RW | GGW | 76.4–109.1 | >13 dB | <0.6 dB |

## 6. Conclusions

Gap waveguide technology makes it possible to have a better tolerance of air gaps. Based on EBG technology, a broadband transition from a GGW to an RW is proposed in this paper. The parameter analysis result is given and the simulation results of the influence of air gaps on transition performance indicate the advantage of the proposed design when facing unwanted air gaps. The back-to-back transition prototype was fabricated and measured. The measured $|S_{11}|$ is better than $-13$ dB and the measured $|S_{21}|$ is better than $-0.6$ dB over 76.4–109.1 GHz. Therefore, the transition structure has good potential in contactless connections and can be applied to mm-wave passive components.

**Author Contributions:** Methodology, X.C. (Xiaohe Cheng); software, Z.L.; formal analysis, Z.L.; validation, Z.L., X.C. (Xiaohe Cheng), Y.Y., J.Y. and X.C. (Xiaodong Chen); investigation, Z.L., X.C. (Xiaohe Cheng) and T.Y.; resources, J.Y.; data curation, Z.L.; writing—original draft preparation, Z.L.; writing—review and editing, X.C. (Xiaohe Cheng), Y.Y, J.Y. and X.C. (Xiaodong Chen); supervision, X.C. (Xiaohe Cheng) and Y.Y.; project administration, Y.Y.; funding acquisition, Y.Y. All authors have read and agreed to the published version of the manuscript.

**Funding:** This work was funded by National Natural Science Foundation of China under grant No. 61474112.

**Conflicts of Interest:** The authors declare that there is no conflict of interests regarding the publication of this article.

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
