# Peer review of "Broadband Transition from Rectangular Waveguide to Groove Gap Waveguide for mm-Wave Contactless Connections"

_electronics, doi:10.3390/electronics9111820_

Round 1
Reviewer 1 Report
Abstract appropriately summarize the paper.
Keywords are appropriate.
- The Introduction is concise.
- Design of the Transition
„In the xz plane, three periods of EBG unit #2 forms the GGW. Its total height is 1.2 mm to match that of WR-10, and the width of the groove in GGW is 2.54 mm.“
Dimensions WR-10 in 2.54x1.27 mm. Why is the EBG unit #2 dimension 1.2 mm?
- Parameter Analysis
„In the back-to-back transition model, z1 ranges from 0.1 to 0.84 mm.“......
„For the best performance the authors choose z1 = 0.84 mm.“
If the selected value is the limit in the range analyzed then it is not selected and the best or optimal value is reached. Namely, it is visible that a further increase of this parameter over 0.84 mm would improve the value of S21.
The explanation of choosing the value of d1 is reasonable and correct but not for choosing z1!
FIG 9.
Even in Figure 9, something is not correct with the dimensions. If we take the height of EBG unit # 2 of 1.2 mm and the distance of 0.05 mm (in Figure 9) then it is 1.25 mm while the dimension of WR-10 is 1.27 mm?
„Therefore, by adding the pins at the interconnection between RW and GGW, a comparatively much better tolerance of air gaps can be achieved.“
It is not clear how this conclusion was reached!
- Simulation and Measured Results
- Conclusions
„Fig. 13, the simulated |S11| is better than -15 dB and |S21| is better than -0.6 dB over the whole 161 W-band. The measured |S11| is better than -13 dB and |S21| is better than -0.6 dB over 76.4-109.1 162 GHz, covering a bandwidth of 35.3%.“
This is a very vaguely written conclusion and should be written more accurately!
Reviewer 2 Report
The paper presents a broadband transition from GGW to RW, based on
EBG technology. The proposed structure may be interesting to the readers. However, there is no clear design method presented.
A quantitative comparison with conventional transitions should be added to the paper.
A better picture of the fabricated structure is also required.
Reviewer 3 Report
In this paper, the authors address a very interesting and timely topic. However, in my opinion there are some points that need to be improved:
1) I am not satisfied by the structure of the Introduction. Even in a coincise way, I think that authors should provide a motivational introduction to the topic of mmWave communications, highlighting the main advantages of such a breakthrough technology. In its actual form, the Introduction delves too quickly into the technical aspects related to the waveguide design, without providing sufficient elements to highlight the usefulness of the addressed topic. For instance, from a quick search through Google Scholar, several application fields of mmWave communications clearly emerge:
5G Networks: S. Okasaka et al, "Proof-of-Concept of a Millimeter-Wave Integrated Heterogeneous Network for 5G Cellular", Sensors 2016, 16, 1362.
Accurate Localization: H. Wymeersch et al, "Low-Complexity Accurate Mmwave Positioning for Single-Antenna Users Based on Angle-of-Departure and Adaptive Beamforming," ICASSP 2020, Barcelona, Spain, 2020, pp. 4866-4870.
Reduce hardware complexity and power consumption: A. N. Uwaechia et al, "On the spectral-efficiency of low-complexity and resolution hybrid precoding and combining transceivers for mmWave MIMO systems", IEEE Access, vol. 7, pp. 109259-109277, Aug. 2019.
I think that these kind of references can be very helpful to introduce an interested reader to the enormous potentials of future mmWave communications, motivating at the same time the importance of the addressed topic.
2) Sec. III: Could you provide a more complete discussion about the effects of wave polarization?
3) In Sec. IV, some additional details about the simulation environment could be helpful to understand how they are conducted and how they relate to the experiments.
Round 2
Reviewer 3 Report
Authors correctly addressed all my comments.